# Multi-Ancestry Transcriptome-Wide Association Studies of Cognitive Function, White Matter Hyperintensity, and Alzheimer’s Disease

**DOI:** 10.3390/ijms26062443

**Published:** 2025-03-09

**Authors:** Dima L. Chaar, Zheng Li, Lulu Shang, Scott M. Ratliff, Thomas H. Mosley, Sharon L. R. Kardia, Wei Zhao, Xiang Zhou, Jennifer A. Smith

**Affiliations:** 1Department of Epidemiology, School of Public Health, University of Michigan, Ann Arbor, MI 48109, USA; dimachaar1@gmail.com (D.L.C.); ratliffs@umich.edu (S.M.R.); skardia@umich.edu (S.L.R.K.); zhaowei@umich.edu (W.Z.); 2Department of Biostatistics, School of Public Health, University of Michigan, Ann Arbor, MI 48109, USA; zlisph@umich.edu (Z.L.); xzhousph@umich.edu (X.Z.); 3Department of Biostatistics, University of Texas MD Anderson Cancer Center, Houston, TX 77030, USA; shanglu@umich.edu; 4Memory Impairment and Neurodegenerative Dementia (MIND) Center, University of Mississippi Medical Center, Jackson, MS 39216, USA; tmosley@umc.edu; 5Survey Research Center, Institute for Social Research, University of Michigan, Ann Arbor, MI 48104, USA

**Keywords:** cognitive function, Alzheimer’s disease, white matter hyperintensity, dementia, vascular dementia, transcriptome-wide association study, multi-ancestry

## Abstract

Genetic variants increase the risk of neurocognitive disorders in later life, including vascular dementia (VaD) and Alzheimer’s disease (AD), but the precise relationships between genetic risk factors and underlying disease etiologies are not well understood. Transcriptome-wide association studies (TWASs) can be leveraged to better characterize the genes and biological pathways underlying genetic influences on disease. To date, almost all existing TWASs on VaD and AD have been conducted using expression studies from individuals of a single genetic ancestry, primarily European. Using the joint likelihood-based inference framework in Multi-ancEstry TRanscriptOme-wide analysis (METRO), we leveraged gene expression data from European ancestry (EA) and African ancestry (AA) samples to identify genes associated with general cognitive function, white matter hyperintensity (WMH), and AD. Regions were fine-mapped using Fine-mapping Of CaUsal gene Sets (FOCUS). We identified 266, 23, 69, and 2 genes associated with general cognitive function, WMH, AD (using EA GWAS summary statistics), and AD (using AA GWAS), respectively (Bonferroni-corrected alpha = *p* < 2.9 × 10^−6^), some of which had been previously identified. Enrichment analysis showed that many of the identified genes were in pathways related to innate immunity, vascular dysfunction, and neuroinflammation. Further, the downregulation of *ICA1L* was associated with a higher WMH and with AD, indicating its potential contribution to overlapping AD and VaD neuropathology. To our knowledge, our study is the first TWAS on cognitive function and neurocognitive disorders that used expression mapping studies for multiple ancestries. This work may expand the benefits of TWASs beyond a single ancestry group and help to identify gene targets for pharmaceuticals or preventative treatments for dementia.

## 1. Introduction

Adult-onset dementia comprises a group of aging-related neurocognitive disorders caused by the gradual degeneration of neurons and the loss of brain function. These changes lead to a decline in cognitive abilities and impairments in daily activities and independent function. In the United States, Alzheimer’s disease (AD), the most common cause of dementia, affects 6.8 million adults aged 65 and older [1]. The second most common form of dementia is vascular dementia (VaD), which often co-occurs with AD and is underdiagnosed [1,2]. VaD is often difficult to distinguish from AD because these diseases share cognitive symptoms, including noticeable impairments in episodic and semantic memory. While AD and VaD often co-occur, each form of dementia has a differing pathophysiology that may precede the illness decades prior.

AD is characterized by the aggregation of amyloid-beta protein and neurofibrillary tangles in brain tissue [3,4], while VaD may be caused by reduced blood flow to the brain as a result of small vessel disease (SVD) or stroke and is commonly seen in people with hypertension [5]. AD is diagnosed based on a battery of memory tests, brain imaging tests to evaluate the degeneration of brain cells, and laboratory tests to assess the presence of amyloid and tau proteins in cerebrospinal fluid [6]. SVD is primarily detected on magnetic resonance imaging (MRI) as white matter hyperintensities (WMHs). It has been hypothesized that vascular and neurodegenerative changes in the brain may interact in ways that increase the likelihood of cognitive impairment. A further challenge in the field is distinguishing between individuals who are aging normally from those with dementia pathology.

A greater understanding of the pathological processes that influence cognitive function in older adults is critical for early intervention during the long preclinical or prodromal phase prior to dementia onset, especially in vulnerable populations [7,8]. For example, African American individuals have a greater burden of and risk for developing dementia compared to non-Hispanic white individuals [9,10,11,12]. Differences in gene expression, which are influenced by both genetic and non-genetic factors, may play a role in shaping racial health disparities in neurological outcomes. However, the underlying molecular and environmental mechanisms that influence gene expression are not fully understood, especially in ancestrally diverse populations. Given the multifactorial and complex nature of dementia, multi-omic data integration across ancestry groups may lend insight into these disparities, allowing for the identification of targets for intervention and treatment in populations that are most at risk [13].

Genome-wide association studies (GWASs) have identified genetic variants associated with cognitive function and dementia; however, most GWAS variants are located in non-coding regions, so their functional consequences are difficult to characterize [14]. Transcriptome-wide association studies (TWASs) utilize gene expression and genetic data to increase the power for identifying gene–trait associations and characterizing the gene expression mechanisms underlying complex diseases. To date, however, few TWASs have been conducted on cognitive or structural brain measures. Further, previous TWASs have primarily been conducted in populations of European ancestry (EA), but these results cannot always be generalized to other genetic ancestries due to differences in allele frequencies, patterns of linkage disequilibrium (LD), and the relationships between single-nucleotide polymorphisms (SNPs) and gene expression among populations [15,16,17,18]. To better identify gene–trait associations in non-European ancestries, it is necessary to incorporate results from recent expression quantitative trait locus (eQTL) mapping studies, which identify genetic variants that explain variations in gene expression levels, conducted in different ancestry groups [19].

Multi-ancEstry TRanscriptOme-wide analysis (METRO) [20] is a TWAS method that uses a joint likelihood-based inference framework to borrow complementary information across multiple ancestries to increase TWAS power. In this study, we employed a multi-ancestry TWAS with the Genetic Epidemiology Network of Arteriopathy (GENOA) dataset (1032 African ancestry (AA) and 801 EA individuals) to identify genes associated with general cognitive function [21], white matter hyperintensity [22], and AD [23,24]. We then examined the contributions of different ancestry-dependent gene expression profiles to the gene–trait associations. Obtaining greater knowledge of the underlying molecular mechanisms of dementia that are generalizable to both EA and AA individuals is a critical step in evaluating potential causal variants and genes that could be targeted for pharmaceutical development.

## 2. Results

In Table 1, we provide descriptive statistics for the samples used in the eQTL mapping study (e.g., 1032 AA and 801 EA individuals from GENOA) and the four input GWASs [21,22,23]. The GENOA eQTL study included participants with a mean age of 56.9 (SD = 10.0) years. More than half of the participants were female (65.6%). The mean age of the participants was 56.9 (SD = 7.8) years in the general cognitive function GWAS [21], and 64.2 years in the WMH GWAS [22]. In the AD GWAS in the EA population [23], the mean age was 73.6 (SD = 8.1) years for cases and 67.9 (SD = 8.6) years for controls. In the AD GWAS in the AA population [24], the mean age was 74.2 (SD = 13.6) years for all participants.

Using METRO, we identified 602 genes associated with general cognitive function, 45 genes associated with WMH, 231 genes associated with AD (EA GWAS), and 9 genes associated with AD (AA GWAS) that were significant at the Bonferroni-corrected alpha level (*p* < 2.90 × 10^−6^; Figure 1, Appendix A).

The genomic inflation factors for the TWAS *p*-values ranged from 1.00 to 2.55 (Figure 2). Among the three neurocognitive outcomes, prior to fine-mapping, the METRO TWAS identified the *ICA1L* gene overlapping between WMH (α = −0.27; *p* = 5.44 × 10^−12^) and AD (from EA GWAS; α = −0.06; *p* = 2.68 × 10^−6^); the *FMNL1* gene overlapping between WMH (α = 0.54; *p* = 1.39 × 10^−8^) and general cognitive function (α = 0.17; *p* = 2.15 × 10^−6^); and 22 genes enriched in AD-related pathways and functions overlapping between general cognitive function and AD (from EA GWAS) (Figure 3a and Appendix A). After fine-mapping, the only overlapping gene that remained was *ICA1L* between WMH and AD (Figure 3b). The METRO TWAS for AD (AA GWAS) identified nine genes overlapping with those identified in AD (EA GWAS); however, following fine-mapping, only *TOMM40* overlapped between the two AD TWASs (EA TWAS: α = −19.78; *p* = 4.77 × 10^−39^; AA TWAS: α = 10.02; *p* = 1.15 × 10^−80^; Figure 3b).

For all identified genes, we also examined the contribution weights of expression prediction models for the EA and AA ancestries, prior to fine-mapping (*p* < 2.90 × 10^−6^; Figure 4). For the WMH TWAS, a significantly higher proportion of genes had a greater contribution from EA weights than AA weights (65.2% vs. 34.8%). This is consistent with Li et al. (2022) [20], who found that gene expression prediction models constructed in the same ancestry as the input GWAS, in this case EA, often had larger contribution weights than those constructed in other ancestries. However, for both general cognitive function and AD (EA and AA GWASs), the contributions from EA and AA weights were similar, which likely increased the power to identify genes relevant to AA.

After fine-mapping, there were 266 genes in the 90%-credible set across 172 different genomic regions for general cognitive function. This gene set included 82 genes that were not previously identified in the SNP-based GWAS results (mapped to the nearest gene) or the gene-based analysis results from Davies et al. (2018) [21] (Figure 5, Appendix A); however, it is likely that some of these genes were in broader genomic regions tagged by the GWAS-identified SNPs. Specifically, there were 126 and 168 overlapping genes between METRO and the SNP-based and gene-based associations from Davies et al. (2018) [21], respectively (Figure 5). The 266 METRO-identified genes were enriched in regulatory pathways involved in protein binding (p_adj_ = 1.17 × 10^−5^), developmental cell growth (p_adj_ = 3.33 × 10^−5^), and the protein metabolic process (p_adj_ = 7.18 × 10^−4^), as well as neurodevelopmental processes such as neuron to neuron synapses (p_adj_ = 1.22 × 10^−3^) and neuron projection (p_adj_ = 7.14 × 10^−3^; Figure 6). The 82 genes that were not previously identified in Davies et al. (2018) [21] were enriched for the positive regulation of biological processes (p_adj_ = 1.77 × 10^−2^), the proteasome activator complex (p_adj_ = 1.00 × 10^−2^), nucleoplasm (p_adj_ = 1.29 × 10^−2^), and chromatin (p_adj_ = 4.71 × 10^−5^; Appendix A).

After fine-mapping, there were 23 genes in the 90%-credible set across 15 genomic regions for WMH, including 12 genes that were not previously identified in the SNP-based GWAS results mapped to the nearest gene or the gene-based analysis results from Sargurupremraj et al. (2020) [22] (Figure 7, Appendix A). Specifically, there were 7 and 12 overlapping genes between METRO and the SNP-based and gene-based associations from Sargurupremraj et al. (2020) [22], respectively (Figure 7). The 23 METRO-identified genes were enriched for zinc finger motif (p_adj_ = 1.27 × 10^−2^), miRNA has-212-5p (p_adj_ = 1.94 × 10^−2^), and the retinal inner plexiform layer (p_adj_ = 3.86 × 10^−2^; Figure 8). The 12 genes associated with WMH that were previously not identified by Sargurupremraj et al. (2020) [22] were enriched for DNA binding domain Zinc Finger Protein 690 (ZNF690; p_adj_ = 2.52 × 10^−3^) and the ClpX protein degradation complex (p_adj_ = 4.97 × 10^−2^; Appendix A).

After fine-mapping, there were 69 genes in the 90%-credible set across 56 genomic regions associated with AD (EA GWAS), including 45 genes that were not previously identified in the SNP-based GWAS results mapped to the nearest gene or the gene prioritization analysis results from Bellenguez et al. (2022) [23] (Figure 9, Appendix A). Specifically, there were 16 and 14 overlapping genes between METRO and the SNP-based and gene prioritization test results from Bellenguez et al. (2022) [23], respectively (Figure 9). The 69 METRO-identified genes were enriched for AD-associated processes, including the regulation of amyloid fibril formation (p_adj_ = 1.87 × 10^−3^), amyloid-beta clearance (p_adj_ = 1.90 × 10^−3^), microglial cell activation (p_adj_ = 5.79 × 10^−3^), the amyloid-beta metabolic process (p_adj_ = 1.07 × 10^−2^), and neurofibrillary tangles (p_adj_ = 2.80 × 10^−4^; Figure 10). The 45 genes associated with AD that were previously not identified by Bellenguez et al. (2022) [23] were enriched for hematopoietic cell lineage (p_adj_ = 1.73 × 10^−3^) and neurofibrillary tangles (p_adj_ = 9.13 × 10^−3^; Appendix A).

We identified 2 genes, *TOMM40* (α = 10.02; *p* = 1.15 × 10^−80^) and *PVRL2* (α = −10.72; *p* = 9.67 × 10^−81^), in the 90%-credible set associated with AD (AA GWAS) (Appendix A). After fine-mapping, none of these genes overlapped with the SNP-based GWAS results mapped to the nearest gene or the gene-based analysis results from Kunkle et al. (2021) [24], since they were both in the broader *APOE* region, which was the only identified gene in Kunkle et al. (2021) [24]. The two METRO-identified genes were enriched for coreceptor-mediated virion attachment to a host (p_adj_ = 4.96 × 10^−2^; Figure 11).

We compared the genes identified by METRO before and after fine-mapping with those identified by the TWASs in Sargurupremraj et al. (2020) [22] and Bellenguez et al. (2022) [23], which used TWAS-Fusion (Figure 12). For WMH, there were 16 and 10 genes identified both by METRO before and after fine-mapping and by the TWAS-Fusion analysis conducted by Sargurupremraj et al. (2020) [22], respectively (Table 2). For AD, there were 24 and 10 genes identified both by METRO before and after fine-mapping and by the TWAS-Fusion followed by FOCUS fine-mapping analysis conducted by Bellenguez et al. (2022) [23] (Table 3). *ICA1L* was the only gene overlapping between the AD and WMH TWAS association results.

## 3. Discussion

While previous studies have identified genes associated with cognitive function, WMH, and AD, there are few TWASs that have utilized genetic and gene expression data from multiple ancestries to increase the power to elucidate the gene–trait associations and molecular mechanisms underlying the etiologies of cognitive function and neurocognitive disorders. Using the METRO method in a GWAS consisting primarily of EA samples followed by FOCUS fine-mapping, we identified 266, 23, and 69 genes associated with general cognitive function, WMH, and AD, respectively, with 82, 12, and 45 of them not previously identified in the original GWAS. In addition, using an AA GWAS, we identified two fine-mapped genes associated with AD, both of which were proximal to *APOE*. Studying the gene expression mechanisms underlying cognitive function, WMH, and dementia using both EA and AA expression data may enhance our understanding of cognitive health prior to and following the onset of dementia and further allow us to generalize the findings from large-scale EA GWASs to other ancestries.

AD and SVD have overlapping features that contribute to dementia neuropathology, including the breakdown of the blood–brain barrier [26] and the presence of small cortical and subcortical infarcts, microbleeds, perivascular spacing, and WMH in brain tissue [27]. After fine-mapping, Islet Cell Autoantigen 1 Like (*ICA1L*) was identified in both the WMH and AD TWASs. This is a highly plausible prioritized gene that is likely to modulate the metabolism of amyloid precursor protein (APP) [23] and increase the risk of AD. *ICA1L* encodes a protein whose expression is activated by type IV collagen and plays a crucial role in myelination [28]. Increased *ICA1L* expression is also associated with a lower risk of AD [29,30,31] and small vessel strokes (SVSs), the acute outcomes of cerebral SVD, which may lead to VaD [32]. Consistent with these studies, our TWAS found that a decreased expression of *ICA1L* is associated with an increased risk of AD and WMH, a subclinical indicator of SVD. Single-cell RNA-sequencing has shown *ICA1L* expression to be enriched in cortical glutamatergic excitatory neurons, which are crucial components in neural development and neuropathology through their roles in cell proliferation, differentiation, survival, neural network formation and cell death [33,34]. *ICA1L* has been examined as a possible drug target for SVD, AD, and other neurodegenerative diseases [32,35]; however, it is not recommended as a prioritized drug at this time due to potential side effects, including an increased risk of coronary artery disease and myocardial infarction, as well as a lower diastolic blood pressure [35]. Nevertheless, future research could validate *ICA1L* as a biomarker through longitudinal studies assessing its levels in cerebrospinal fluid and blood, investigating its ability to cross the blood–brain barrier, and characterizing how it correlates with dementia-related outcomes [36,37,38,39]. For instance, as both SVD and AD are associated with blood–brain barrier dysfunction, *ICA1L*’s protective effect could potentially be mediated through blood–brain-barrier-related mechanisms [36,37,38,39]. As such, further investigation of *ICA1L* and imaging of the blood–brain barrier may allow for a greater understanding of this gene as a therapeutic target. As such, *ICA1L* may contribute to overlapping AD and VaD neuropathology, and it could be a potential target for therapeutics and/or preventative treatments for AD and VaD in the future if adverse events can be reduced.

Our TWAS of AD (from the EA GWAS) identified 45 genes that were not identified in the SNP-based GWAS results mapped to the nearest gene or the gene-based analysis reported in Bellenguez et al. (2022) [23]. The 45 genes were enriched for hematopoietic cell lineages, which are progenitors of red and white blood cells, including those related to immunity (e.g., natural killer cells, T- and B-lymphocytes, and other types of leukocytes) [40,41,42,43,44,45,46]. Our TWAS identified genes that have been previously associated with AD, including *APOE*, *TOMM40*, *APOC4*, *CLU*, *PICALM*, and *CR2*, among others [23,47,48]. While we identified *APOE,* the strongest genetic risk factor for AD in most populations, after fine-mapping, we did not identify *ABCA7*, which confers an equal or even greater risk for AD in AA individuals [49,50,51]. This finding is perhaps not surprising considering that our TWAS was conducted using an EA GWAS, and the strength of association between *ABCA7* and AD is comparatively weaker in EA samples than in AA ones [51]. To identify genes associated with AD risk in AA populations, specifically, it would be beneficial to perform a TWAS utilizing a well-powered AD GWAS in AA individuals.

Our TWAS of AD (from the AA GWAS) [24] identified only *TOMM40* and *PVRL2*, both proximal to *APOE*. *TOMM40* was also identified in our TWAS of AD (from the EA GWAS), as well as other AD GWASs [23,47,48]. *PVRL2* has been associated with metabolic syndrome, diabetic dyslipidemia, and AD [52,53]. One study found that polymorphisms in *PVRL2* interact with variants in *TOMM40* to increase AD risk through pathways related to amyloid-beta metabolism in older Chinese adults [53]. As larger AD GWASs in AA populations become available, we may be able to identify additional genes associated with this disease while leveraging gene expression data from EA and AA samples.

In our AD EA TWAS, we also identified genes associated with other neurological and autoimmune diseases, including Parkinson’s disease (*CYB561* [54] and *SLC25A39* [55]), Crohn’s disease (*ATG16L1* [56]), Amyotrophic lateral sclerosis (*SIGLEC9* [57]), and Riboflavin Transport Deficiency (*SLC52A1* [58]). These diseases have in common the progressive peripheral and cranial degeneration of neurons that impact processes such as voluntary muscle movement, vision, hearing, and sensation. Although not explicitly identified in Bellenguez et al. (2022) [23], we also identified genes that were associated with AD in other studies, including *RIN3* that is implicated in tau-mediated pathology, the *MS4A* (*4A* and *6A*) locus associated with mast cell activation, *TP53INP1* and *ZYX* that have been linked to myeloid enhancer activity [59], and *APOC4*, which is located proximal to *APOE* [60]. We also identified additional genes involved in B cell autoimmunity (*HLA-DQA2* [61,62] and *CSTF1* [63]), neurodegenerative processes (*SUPT4H1* [64], *C6orf10* [65], *IKZF1* [66], and *DEDD* [67]), and neuronal growth (*IKZF1* [66] and *STYX* [68]). Our findings support the hypothesis that the chronic activation of immune cells resident in the brain and peripheral nervous system appears to play a critical role in neuroinflammatory responses that drive the progression of neurodegeneration in AD [69]. Further, consistent with findings that AD and VaD often co-exist, our AD TWAS identified genes that were associated with lacunar and ischemic strokes, as well as cerebral small vessel disease in other studies, including *SLC39A13* [70], *RAPSN* [70], *MAF1* [71], and *MME* [72,73].

Although our WMH TWAS identified 12 genes that were not included in the SNP-based GWAS results mapped to the nearest gene or the gene-based analysis reported in Sargurupremraj et al. (2020) [22], other studies found associations between *MAP1LC3B* [74], *ARMS2* [75,76], and *HTRA1* [70] with ischemic stroke, lacunar stroke, and cerebral SVD. The WMH TWAS also identified genes associated with AD (*ARMS2*) [77], atrial fibrillation (*NEURL* [78] and *GJC1* [79]), innate immunity (*EFTUD2* [80]), and apoptosis and neurodevelopment (*PDCD7* [81], *FBXO31* [82], and *ClpX* [83]). The 12 unique genes identified for WMH were enriched for DNA-binding domain Zinc Finger Protein 690 [84], which plays an essential role in gene regulation, transcription, and various cellular processes, and the ClpX protein degradation complex [85], which maintains protein homeostasis. Our findings are consistent with studies that have shown neuroinflammation to be an immunological cascade reaction by glial cells of the central nervous system, where innate immunity resides.

While our TWAS for general cognitive function did not show overlapping genes between the TWAS for AD and VaD, we identified genes associated with general cognitive function that were not explicitly identified by Davies et al. (2018) [21] which were associated with pre-clinical AD and VaD risk factors, including cardiovascular diseases, immunity, and Alzheimer’s neuropathology. Our TWAS also identified genes previously associated with cognitive domains, neuropathology, and psychiatric illness, including reading-related skills and neural structures (*SEMA6D* [86] and *SETBP1* [87]), working memory tasks (*CDH13* [88]), and Schizophrenia (*HP* [89,90], *C18orf1* [91], and *TMEM180* [92]). There are likely also distinct gene expression mechanisms that differentiate cognitive function and normal age-related brain changes from pathways related to dementia. Individuals who never develop dementia or significant cognitive decline still experience brain deterioration in normal aging, which includes gray and white matter loss and ventricular enlargement that is accompanied by memory decline [93]. Further, previous GWASs for general cognitive function and AD have shown few overlapping loci [21,94]. In addition, studies of older individuals who are cognitively “resilient” with intact cognitive function, despite the presence of AD neuropathology, have found the genetic architecture of cognitive resilience to be distinct from that of AD [95]. As such, relatively little is known about the pathways underlying cognitive aging in those without dementia. Thus, studying gene expression mechanisms that affect general cognitive function before the development of dementia may shed light on cognitive aging without dementia.

Interestingly, while our study’s TWASs for AD (EA and AA), WMH, and general cognitive function did not reveal overlapping genes across all four outcomes, we did identify shared underlying metabolic pathways and signaling processes. Specifically, we identified genes related to insulin signaling, whose dysfunction has been implicated in synaptic dysfunction, neuroinflammation, and AD pathologies. Insulin, an anabolic hormone that regulates glucose metabolism, is able to freely cross the blood–brain barrier from the circulatory system and is known to support cognition and enhance the outgrowth of neurons [96]. Studies have shown that insulin resistance, a hallmark of type 2 diabetes, can lead to cognitive decline and is associated with the development of AD [97]. METRO-identified genes for AD (EA and AA), WMH, and general cognitive function have been implicated in insulin resistance and metabolic dysfunction. For instance, for AD, *TOMM40* has been associated with mitochondrial dysfunction and neuroinflammation, which can exacerbate insulin resistance [98], while *APOE*, particularly the *APOE*-ε4 allele, is a well-established risk factor for late-onset AD and has been shown to interfere with insulin receptor signaling, trapping insulin receptors in endosomes and impairing their function [99]. For WMH, *WBP2* has been shown to play a role in the endocrine system, suggesting its involvement in insulin signaling, lipid metabolism, and other metabolic functions [100,101]. Lastly, for general cognitive function, *FOXO6* integrates insulin signaling with gluconeogenesis in the liver [102,103] and *INS* encodes the peptide hormone insulin, which plays a crucial role in regulating the carbohydrate and lipid metabolisms [104]. Collectively, our findings highlight the complex interplay between insulin signaling, metabolic dysfunction, and cognitive health, underscoring the potential for therapeutic strategies impacting these pathways to mitigate the impacts of cognitive aging and AD.

We also compared genes identified by METRO after fine-mapping with those identified by TWAS-Fusion in Sargurupremraj et al. (2020) [22] and Bellenguez et al. (2022) [23]. Among the 92 genes associated with WMH in Sargurupremraj et al. (2020) [22] and 23 genes identified by METRO, 10 genes overlapped. We note that Sargurupremraj et al. (2020) [22] did not perform the fine-mapping of their TWAS results, which is likely why we identified substantially fewer genes. There were also 10 overlapping genes among the 66 genes associated with AD in Bellenguez et al. (2022) [23] and 69 genes identified by METRO. For both TWAS comparisons, a relatively small number of genes overlapped, likely due to differences in eQTL prediction modeling. Sargurupremraj et al. (2020) [22] and Bellenguez et al. (2022) [23] used eQTL data from brain tissue, while we used eQTL data from transformed beta lymphocytes in blood tissue. While brain tissue is more relevant to WMH and AD phenotypes, blood cells do touch every cell bed that affects the brain and are related to chronic inflammation, immunity, and oxidative stress, which are linked to cognitive performance and dementia. TWAS results from blood tissue from multiple ancestries provide complementary information to those reported in GWASs.

Several limitations in the present study should be noted. First, our gene expression levels were measured using transformed B-lymphocytes from immortalized cell lines in GENOA. While transformed B-lymphocytes are a convenient source of DNA from blood tissue, we lack eQTL data for the tissues that may be most relevant for AD and WMH (e.g., brain tissue, small brain vessels, and microglia). However, B-lymphocytes provide a unique and efficient way to examine the functional effects of genetic variations on gene expression while minimizing environmental influences [105]. Second, METRO follows the standard TWAS approach of analyzing one gene at a time. Since genes residing in the same genomic region may share eQTLs or contain eQTL SNPs that are in LD with each other, the TWAS test statistics of genes in the same region may be highly correlated. To this end, it may be challenging to identify the truly biologically relevant genes among them [25,106]. As such, we paired METRO with FOCUS to allow us to narrow down the list of potential causal genes for AD, VaD, and cognitive decline [25,107]. In addition, we primarily utilized EA GWASs that were publicly available with large sample sizes for general cognitive function, WMH, and AD. As expected, the gene expression prediction models constructed in the same ancestry as the GWAS (EA) tended to have larger contribution weights than those for AA. While we conducted a TWAS of AD in AA individuals, the sample size of the AA GWAS likely did not allow us to properly power our TWAS. As such, a future direction would be to conduct TWASs of these traits using summary statistics from well-powered GWASs with AA ancestry or multiple ancestries as they become available. Lastly, while our study focused on the relationship between genetically regulated expression and neurocognitive outcomes, research has shown that environmental factors, individual lifestyle factors, and epigenetic factors interact with and influence gene expression, which also affects neurocognitive differences [108,109,110,111,112,113,114,115,116,117,118,119,120,121,122,123,124,125,126,127,128,129,130]. For example, to investigate the underlying biological pathways of cognitive aging, recent investigations by our group and others have begun to elucidate the underlying pathways of cognitive aging by examining these complex relationships [108,117,131,132,133,134]. Future studies would benefit from the incorporation of information on environmental and individual lifestyle factors, as well as multi-omic data, across ancestry groups to better understand ancestry-specific gene expression differences.

While a full characterization of the genes associated with neurocognitive traits is beyond the scope of this study, previous studies have functionally characterized and experimentally validated some of these identified genes. For instance, studies of postmortem brain tissue from individuals carrying an *APOE-ε4* allele [135,136], as well as investigations using *TREM2* knockout human cell lines [137,138], have shown an earlier onset and more abundant amyloid pathology in such brain tissue. Additionally, genes like *CALCRL* and *EFEMP1* have shown potential neuroprotective effects and associations with neuroinflammation in AD-related conditions [139,140,141,142]. Further functional validation of the identified genes in future studies would allow for an enhanced biological interpretation of our findings. The inability to replicate or validate significant genes using additional datasets poses a challenge to the generalizability of our findings. Future studies incorporating larger and more diverse cohorts, as well as integrating multi-omic data, will be crucial for overcoming these limitations and providing a more comprehensive understanding of the genetic factors influencing neurocognitive traits.

Our study also has notable strengths. To the best of our knowledge, our study is the first TWAS using expression mapping studies in multiple ancestries (EA and AA) to identify genes associated with cognitive function and neurocognitive disorders. By conducting a TWAS, we identified putatively causal genes whose expression levels were associated with AD, VaD, and general cognitive function risk through genetically regulated expression, leveraging eQTLs as instrumental variables [20,143,144,145,146,147]. As such, TWASs can be viewed as a form of two-sample two-stage least squares analysis, which are closely related to Mendelian randomization but specifically tailored for gene expression studies [145]. By leveraging the complementary information in gene expression prediction models constructed in EA and AA populations, as well as the uncertainty in SNP prediction weights, we were able to conduct a highly powered TWAS to identify important gene–trait associations and gene expression mechanisms related to the innate immunity, vascular dysfunction, and neuroinflammation underlying AD, VaD, and general cognitive function. Using METRO, we were also able to estimate the ancestry contribution weights for specific genes and identify the extent to which a gene in EA or AA may contribute to the trait. However, it is noteworthy that the larger contributions of the expression prediction models in the same ancestry as the GWAS (primarily EA in this study) may allow for better predictive performance in the same ancestry. The METRO framework provides a higher statistical power for TWASs as compared to ancestry-specific approaches, which we consider to be crucial in GWAS applications to underrepresented populations with small sample sizes. In Li et al. [20], METRO was compared to various alternative methods, including ancestry-specific TWASs, TWAS meta-analysis, and a combination test that aggregates *p*-values from ancestry-specific TWASs. The comparison considered both Bayesian and Frequentist imputation techniques, where the Bayesian sparse linear mixed model (BSLMM) and the elastic net model were used for gene expression prediction. In both simulation studies and real data applications, METRO provided a well-controlled type-I error and was able to leverage information across multiple gene expression studies to achieve a higher statistical power than the compared methods. These key features of METRO make it well-suited for our multi-ancestry TWAS analysis. We also conducted FOCUS fine-mapping to narrow down a list of putatively causal genes among multiple significant genes in a region. Our results suggest that there are similar pathways that contribute to healthy cognitive aging and the progression of dementia, as well as distinct pathways that are unique to each neuropathology. By understanding overlapping and unique genes and the gene expression mechanisms associated with each outcome, we may identify possible targets for the prevention of and/or treatments for cognitive aging and dementia.

## 4. Materials and Methods

### 4.1. Sample

#### The Genetic Epidemiology Network of Arteriopathy (GENOA)

The GENOA study is a community-based longitudinal study aimed at examining the genetic effects of hypertension and related target organ damage [148]. EA (those who self-identified as non-Hispanic white) and AA (those who self-identified as African American) hypertensive sibships were recruited if at least 2 siblings were clinically diagnosed with hypertension before age 60. All other siblings were invited to participate, regardless of their hypertension status. The exclusion criteria included secondary hypertension, alcoholism or drug abuse, pregnancy, insulin-dependent diabetes mellitus, active malignancy, or serum creatinine levels of >2.5 mg/dL. In Phase I (1996–2001), 1854 AA participants (Jackson, MS, USA) and 1583 EA participants (Rochester, MN, USA) were recruited [148]. In Phase II (2000–2004), 1482 AA and 1239 EA participants were successfully followed up, and their potential target organ damage from hypertension was measured. Demographics, medical history, clinical characteristics, information on medication use, and blood samples were collected in each phase. After data cleaning and quality control, a total of 1032 AA and 801 EA samples with genotype and gene expression data were available for analysis. Written informed consent was obtained from all participants, and approval was granted by participating institutional review boards (University of Michigan, University of Mississippi Medical Center, and Mayo Clinic).

### 4.2. Measures

#### 4.2.1. Genetic Data

AA and EA blood samples were genotyped using the Affymetrix^®^ Genome-Wide Human SNP Array 6.0 (Santa Clara, CA, USA) or the Illumina 1M-Duo 3.0 BeadChip (San Diego, CA, USA). We followed the procedures outlined by Shang et al. [18] for data processing. For each platform, samples and SNPs with a call rate of <95%, samples with a mismatched sex, and duplicate samples were excluded. After removing outliers identified from a genetic principal component analysis, there were 1599 AA and 1464 EA samples with available genotype data. Imputation was performed using the Segmented HAPlotype Estimation & Imputation Tool (SHAPEIT) v.2.r [149] and IMPUTE v.2 [150] using the 1000 Genomes project phase I integrated variant set release (v.3) in NCBI build 37 (hg19) coordinates (released in March 2012). Imputation for each genotyping platform was performed separately and then combined. The final set of genotype data included 30,022,375 and 26,079,446 genetic variants for AA and EA, respectively. After removing genetic variants with an MAF of ≤0.01, an imputation quality score (INFO score) of ≤0.4 in any platform-based imputation, and indels, a total of 13,793,193 SNPs in AA and 7727,215 SNPs in EA were available for analysis. We used the GENESIS package [151] in R version 4.3.3 [152] to infer the population structure in the analytic sample and the PC-AiR function was used to extract the first five genotype principal components (PCs) by ancestry group, which were subsequently used to adjust for population structure.

#### 4.2.2. Gene Expression Data

Gene expression levels were measured from Epstein–Barr virus (EBV)-transformed B-lymphoblastoid cell lines (LCLs) created from blood samples from a subset of GENOA AA (n = 1233) and EA (n = 919) individuals. The gene expression levels of the AA samples were measured using the Affymetrix Human Transcriptome Array 2.0, while the gene expression levels of the EA samples were measured using Affymetrix Human Exon 1.0 ST Array. We processed the data from the AA and EA samples separately, following the procedures outlined in Shang et al. [18]. Briefly, the Affymetrix Expression Console was used for quality control, and all array images passed visual inspection. Affymetrix Power Tool software (version 1.4.1) was used to process raw-intensity data [153]. We normalized Affymetrix CEL files using the Robust Multichip Average (RMA) algorithm, including background correction, quantile normalization, log_2_-transformation, and probe set summarization [154]. Linearity was also maintained using GC correction (GCCN), signal space transformation (SST), and gain lock (value = 0.75). We used the Brainarray custom CDF [155] version 19 to map the probes to genes, specifically removing probes with non-unique matching cDNA/EST sequences that could be assigned to more than one gene cluster. As a result, the gene expression data processed through the custom CDF are expected to be free of mapability issues; however, alignment bias may still exist due to genetic variation, errors in the reference genome, and other complications [156]. After mapping, Combat was used to remove batch effects [157]. Gene expression data were then quantile-normalized across genes.

#### 4.2.3. GWAS Summary Statistics

We used summary statistics from GWASs for general cognitive function [21], WMH [22], AD in EA [23], and AD in AA [24] as inputs for METRO. Three of the GWASs, Davies et al. (2018) [21], Sargurupremraj et al. (2020) [22], and Bellenguez et al. (2022) [23], were selected because they are the largest meta-analyses to date with publicly available summary statistics; however, we note that all three were conducted primarily with EA samples. We also selected the Kunkle et al. (2021) [24] GWAS because it is the largest meta-analysis to date with public available summary statistics for primarily AA samples. Below, we describe each GWAS and also provide information about the corresponding TWAS analyses that were reported in two of the input GWASs (WMH [22] and AD in EA [23]), which use the same GWAS summary statistics as our analysis but different gene expression data.

##### General Cognitive Function

We obtained the GWAS summary statistics for general cognitive function from a meta-analysis by Davies et al. (2018) [21], which includes the Cohorts for Heart and Aging Research in Genomic Epidemiology (CHARGE), the Cognitive Genomics Consortium (COGENT) consortia, and the UK Biobank (UKB; Table 1) [21]. This study included 300,486 EA individuals with ages between 16 and 102 years from 57 population-based cohorts. This is the largest available GWAS for general cognitive function, and there are currently no large-scale GWASs available in non-EA populations. General cognitive function was constructed from a number of cognitive tasks. Each cohort was required to have tasks that tested at least three different cognitive domains. PC analysis was performed on the cognitive test scores within each cohort, and the first unrotated component was extracted as the measure of general cognitive function. The models performed within each cohort were adjusted for age, sex, and population stratification. The exclusion criteria included clinical stroke (including self-reported stroke) or prevalent dementia.

##### White Matter Hyperintensity

We obtained the GWAS summary statistics for WMH from a meta-analysis conducted by Sargurupremraj et al. (2020) that included 48,454 EA and 2516 AA individuals with mean age of 66.0 (SD = 7.5) years from 23 population-based studies from the CHARGE consortium and UKB (Table 1) [22]. We obtained publicly available GWAS summary statistics from only EA individuals. Only summary statistics for EA samples are publicly available for this GWAS. WMH was measured from MRI scans obtained from scanners with field strengths ranging from 1.5 to 3.0 Tesla and interpreted using a standardized protocol blinded to clinical and demographic features. In addition to T1- and T2-weighted scans, some cohorts included fluid-attenuated inversion recovery (FLAIR) and/or proton density (PD) sequences to measure WMH from cerebrospinal fluid. WMH volume measures were inverse normal transformed, and models were adjusted for sex, age, genetic PCs, and intracranial volume (ICV). The exclusion criteria included a history of stroke or other pathologies that influence the measurement of WMH at the time of MRI.

To functionally characterize and prioritize individual WMH genomic risk loci, Sargurupremraj et al. [22] (2020) conducted a TWAS using TWAS-Fusion [144] with summary statistics from the WMH SNP main effects (EA only) analysis and weights from gene expression reference panels from blood (Netherlands Twin Registry; Young Finns Study), arterial (Genotype-Tissue Expression, GTEx), brain (GTEx, CommonMind Consortium), and peripheral nerve tissue (GTEx). This study did not perform fine-mapping following the TWAS analysis.

##### Alzheimer’s Disease (GWAS in EA)

We obtained the EA GWAS summary statistics for Alzheimer’s disease from a stage I meta-analysis by Bellenguez et al. (2022) [23] that included EA samples from the European Alzheimer and Dementia Biobank (EADB), GR@ACE, EADI, GERAD/PERADES, DemGene, Bonn, the Rotterdam study, CCHS study, NxC, and the UKB (Table 1) [23]. The meta-analysis was performed on 39,106 clinically diagnosed AD cases, 46,828 proxy AD and related dementia (ADD) cases, and 401,577 controls. AD cases were clinically diagnosed in all cohorts except UKB, where individuals were identified as proxy ADD cases if their parents had dementia. Participants without a clinical diagnosis of AD, or those without any family history of dementia, were used as controls. Models performed within each cohort were adjusted for PCs and genotyping centers when necessary.

To examine the downstream effects of new AD-associated variants on molecular phenotypes in various AD-relevant tissues, Bellenguez et al. (2022) [23] conducted a TWAS with stage I AD GWAS results. The TWAS was performed by training functional expression and splicing reference panels based on the Accelerating Medicines Partnership (AMP) AD bulk brain and EADB lymphoblastoid cell line (LCL) cohorts, while leveraging pre-calculated reference panel weights [158] for the GTEx dataset [159] in tissues and cells of interest. TWAS associations were then fine-mapped using Fine-mapping Of CaUsal gene Sets (FOCUS) [25].

##### Alzheimer’s Disease (GWAS in AA)

We also acquired the AA GWAS summary statistics for Alzheimer’s disease from a meta-analysis by Kunkle et al. (2021) [24] that included individuals of African American ancestry from 15 cohort studies from the AD Genetics Consortium (ADGC; Table 1). The meta-analysis was performed on 2748 clinically diagnosed AD cases and 5222 controls with a mean age of 74.2 years (SD = 13.6). The models performed within each cohort were adjusted for age, sex, and genetic PCs.

### 4.3. Statistical Methods

#### 4.3.1. Multi-Ancestry Transcriptome-Wide Association Study

Using the Multi-ancEstry TRanscriptOme-wide (METRO) analysis [20], we conducted a high-powered TWAS with calibrated type I error control to identify the key gene–trait associations and gene expression mechanisms underlying general cognitive function, WMH, and AD. Since gene expression prediction models constructed in different ancestries may contain complementary information, even when the input GWAS was conducted in a single ancestry [20], we used METRO to model gene expressions from EA and AA simultaneously. METRO uses a joint likelihood framework that accounts for SNP effect size heterogeneity and LD differences across ancestries. The framework selectively upweights information from the ancestry that has a greater certainty in the gene expression prediction model, increasing power and allowing for characterization of the relative contributions of each ancestry to the TWAS results.

METRO is described in Li et al. [20]. Briefly, each gene is examined separately using gene expression data from *M* different genetic ancestries. ***Z_m_*** is the *n_m_*-vector of gene expression measurements on *n_m_* individuals in the *m*th ancestry with *m* ∈ {1, …, M}. For the gene of interest, all *cis*-SNPs (*p*), which are in potential linkage disequilibrium (LD) with each other, are extracted as predictors for gene expression. ***G_m_*** is denoted as the *n_m_* × *p* genotype matrix for these *cis*-SNPs. Besides the gene expression data, we also use GWAS summary statistics from *n* individuals for an outcome trait of interest. ***γ*** is the *n*-vector of outcome measurements in the GWAS data and ***G*** is the corresponding *n* × *p* genotype matrix on the same set of *p cis*-SNPs. The expression vector ***z_m_***, the outcome vector ***γ,*** and each column of the genotype matrixes are centered and standardized. ***G_m_*** and ***G*** have a mean of zero and variance of one. For each TWAS, we use GWAS summary statistics in the form of marginal *z*-scores and an SNP–SNP correlation (LD) matrix estimated with genotype data from GENOA that corresponds with the ancestry of the GWAS (EA or AA). The following equations describe the relationships between the SNPs, gene expression, and the outcome:***z_m_
****= **G**_m_**β**_m +_
**ε**_m_*,(1)***γ = ****α(**Gβ**) + **ε**_y_*,(2)

Equation (1) describes the relationship between gene expression and the *cis-*SNP genotypes in the gene expression study in GENOA for the *m*th ancestry (EA or AA). *β_m_* is a *p* vector of the *cis*-SNP effects on the gene expression in the *m*th ancestry and *ε_m_* is an *n_m_*-vector of residual errors, with each element following an independent and normal distribution N(0, *σ*^2^*_m_*) with an ancestry-specific variance *σ*^2^*_m_*. Equation (2) describes the relationship between the genetically regulated gene expression (GReX), calculated from estimated SNP prediction weights, and the outcome trait (general cognitive function, WMH, or AD) from the GWAS. There, *Gβ* denotes an *n*-vector of the GReX constructed for the GWAS individuals, where *β* = Σ*_m_ w_m_β_m_* is a *p*-vector of SNP effects on the gene expression in the GWAS data, where the weights Σ*^M^_m_* _= 1_ *w_m_* = 1 and *w_m_* ≥ 0. The alpha value (*α*) is the effect of the GReX constructed for the GWAS individuals on the outcome trait and *ε_y_* is an *n_m_*-vector of residual errors, with each element following an independent and normal distribution N(0, *σ*^2^*_m_*). Both equations, specified based on separate studies, are connected through the predictive SNP effects on the gene expression (*β_m_* and *β*). A key assumption made is that the SNP effects on the gene expression in the GWAS, *β*, can be expressed as a weighted summation of the SNP effects on gene expression in the expression studies conducted across ancestries.

We derived the overall GReX effect *α* and the contribution weight of each ancestry (w_1_ for AA and w_2_ for EA) to infer the extent and contribution of the two genetic ancestries in informing the GreX–trait association. The joint model defined in Equations (1) and (2) allows us to borrow association strengths across multiple ancestries to enable powerful inferences of GreX–trait associations for general cognitive function, WMH, and AD. We declared the gene to be significant if the *p*-value was below the corresponding Bonferroni-corrected threshold for the number of tested genes (*p* < 0.05/17, 238 = 2.90 × 10^−6^). Manhattan plots and quantile–quantile (Q-Q) plots were generated using the *qqman* [160] R package.

#### 4.3.2. Fine-Mapping Analysis

Since genes residing in the same genomic region may share eQTLs or contain eQTL SNPs in LD with each other, TWAS test statistics for genes in the same region can be highly correlated, making it difficult to identify the true biologically relevant genes among them. To prioritize the putatively causal genes identified by METRO for general cognitive function, WMH, and AD, we conducted TWAS fine-mapping using FOCUS (Fine-mapping Of CaUsal gene Sets) [25]. To identify a genomic region with at least one significant gene detected by METRO, we obtained a set of independent, non-overlapping genomic regions, or LD blocks, using Ldetect [161]. In each analyzed genomic block, using a standard Bayesian approach, we assigned a posterior inclusion probability (PIP) for each gene to be causal, given the observed TWAS statistics. We used gene-level Z scores, created from *p*-values using the inverse cumulative distribution function (CDF) of a standard normal distribution, as inputs for FOCUS. We then ranked the PIPs and computed the 90%-credible set that contained the causal gene with a 90% probability. In the FOCUS analysis, a null model which assumed none of the genes in the region were causally associated with the trait was also considered as a possible outcome and may be included in the credible set. Through fine-mapping, we narrowed down significantly associated genes identified by METRO to a shorter list of putatively true associations.

#### 4.3.3. Characterization of Identified Genes

To interpret our TWAS findings, both before and after fine-mapping, we further examined whether the genes identified by METRO overlapped with those previously identified by their corresponding input GWASs. We created a set of Venn diagrams of overlapping genes identified using METRO with those from the SNP-based GWAS association results [21,22,23,24] mapped to the nearest gene using the *VennDiagram* R package [162]. We then constructed a second set of Venn diagrams showing the overlapping genes identified using METRO with genes identified by gene-based association analyses in each of the input GWASs. The gene-based analyses were conducted using MAGMA [163] (general cognitive function [21], WMH [22], and AD (AA GWAS) [24]) or gene prioritization tests (AD (EA GWAS) [23]). Finally, we created a set of Venn diagrams comparing the genes identified using METRO with those identified in the TWASs that were conducted as part of the WMH [22] and AD (EA GWAS) [23] input GWASs. We used the *geneSynonym* R package [164] to ensure that genes named differently across studies were captured.

#### 4.3.4. Functional Enrichment Analysis

To characterize the biological function of the genes identified by METRO for general cognitive function, WMH, and AD, we performed gene set enrichment analysis. Specifically, we used the g:GOSt [165] tool on the web software g:Profiler (version e112_eg59_p19_25aa4782) and mapped the genes to known functional informational sources, including Gene Ontology (GO): molecular function (MF), GO: biological process (BP), GO: cellular component (CC), Kyoto Encyclopedia of Genes and Genomes (KEGG), Reactome (REAC), WikiPathways (WP), Transfac (TF), MiRTarBase (MIRNA), Human Protein Atlas (HPA), CORUM protein complexes, and Human Phenotype Ontology (HP). In this analysis, we used the default option g:SCS method (Set Counts and Sizes) in g:Profiler for multiple testing correction and presented the pathways identified with an adjusted *p*-value < 0.05. Driver terms in GO are highlighted using a two-stage algorithm for filtering GO enrichment results, providing a more efficient and reliable approach compared to traditional clustering methods. This feature groups significant terms into sub-ontologies based on their relations, and the second stage identifies leading gene sets that give rise to other significant functions in the same group of terms. This method uses a greedy search strategy that recalculates hypergeometric *p*-values and results in the consideration of multiple leading terms in a component, rather than selection of terms with the highest significance level.

## 5. Conclusions

In the present study, we conducted a multi-ancestry TWAS in EA and AA populations to identify genes associated with general cognitive function, WMH, and AD. We identified genes associated with innate immunity, vascular dysfunction, and neuroinflammation. The WMH and AD TWASs also indicated that the downregulation of *ICA1L* may contribute to overlapping AD and VaD neuropathology. To our knowledge, this study is the first TWAS analysis using expression mapping studies in multiple ancestries to increase the power to identify genes associated with cognitive function and neurocognitive disorders, which may help to identify gene targets for pharmaceutical or preventative treatment for dementia.

## Figures and Tables

**Figure 1 ijms-26-02443-f001:**
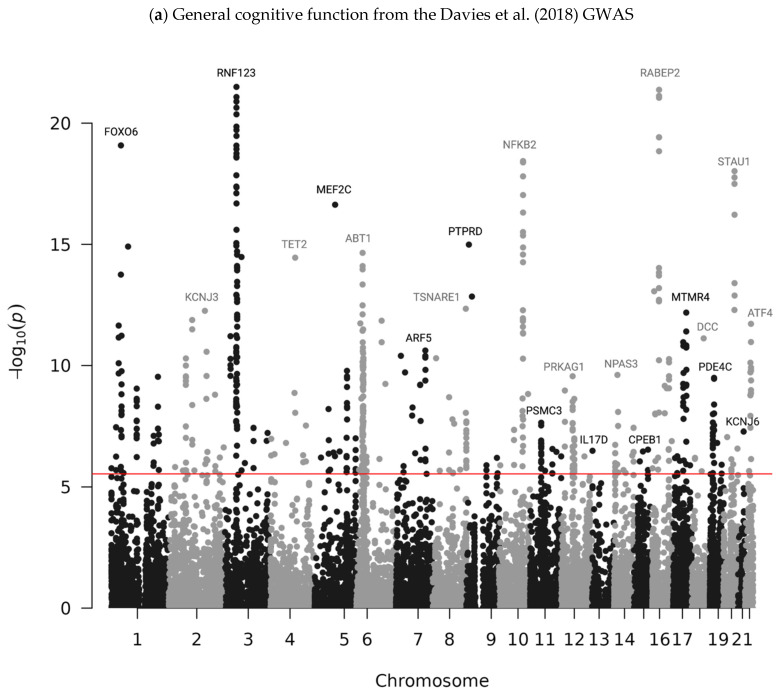
Manhattan plots of −log_10_ *p*-values for gene–trait associations in METRO. Manhattan plots of the association between genes and (**a**) general cognitive function using summary statistics from Davies et al. (2018) [21], (**b**) white matter hyperintensity from Sargurupremraj et al. (2020) [22], (**c**) Alzheimer’s disease from Bellenguez et al. (2022) (EA GWAS sample) [23], and (**d**) Alzheimer’s disease from Kunkle et al. (2021) (AA GWAS sample) [24] using GENOA gene expression data. The red line indicates significance after Bonferroni correction (*p* < 2.90 × 10^−6^).

**Figure 2 ijms-26-02443-f002:**
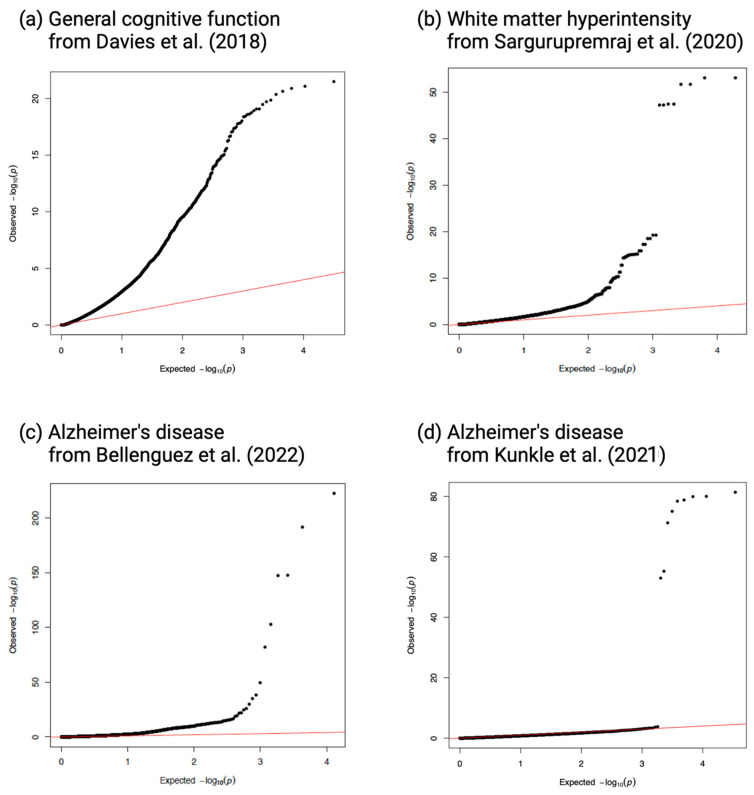
Quantile–quantile (Q-Q) plots of −log_10_ *p*-values for gene-trait associations in METRO. Q–Q plots of the associations between genes and (**a**) general cognitive function (λ = 2.55) using summary statistics from Davies et al. (2018) [21], (**b**) white matter hyperintensity (λ = 1.45) from Sargurupremraj et al. (2020) [22], (**c**) Alzheimer’s disease (λ = 2.09) from Bellenguez et al. (2022) (EA GWAS sample) [23], and (**d**) Alzheimer’s disease (λ = 1.0) from Kunkle et al. (2021) (AA GWAS sample) [24] using GENOA gene expression data.

**Figure 3 ijms-26-02443-f003:**
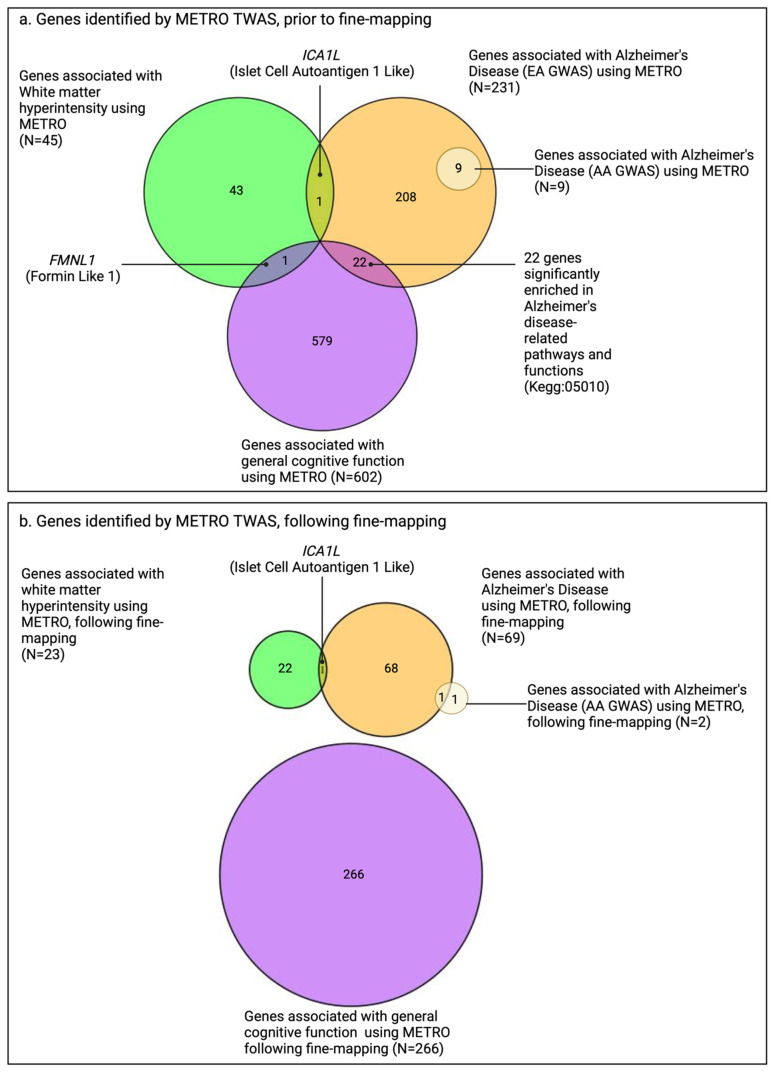
Venn diagrams comparing number of genes associated with general cognitive function, white matter hyperintensity, and Alzheimer’s disease (AD) in European ancestry (EA) and AD in African ancestry (AA) using METRO, prior to and following FOCUS fine-mapping. Venn diagrams comparing the number of genes associated with general cognitive function (purple; N = 266 genes), white matter hyperintensity (WMH; green; N = 23 genes), and Alzheimer’s disease (AD) in EA (orange; N = 69 genes) and AD in AA (yellow; N = 2 genes), (**a**) prior to fine-mapping and (**b**) following FOCUS [25] fine-mapping using METRO and GENOA expression data after Bonferroni correction (*p* < 2.90 × 10^−6^), with GWAS summary statistics obtained from the Davies et al. (2018) [21], Sargurupremraj et al. (2020) [22], Bellenguez et al. (2022) [23], and Kunkle et al. (2021) [24].

**Figure 4 ijms-26-02443-f004:**
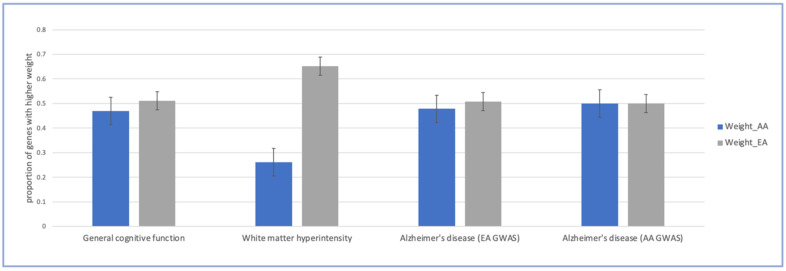
Contribution weights of expression prediction models across all significant fine-mapped genes identified by METRO. Barplots of general cognitive function, white matter hyperintensity, and Alzheimer’s disease (AD) in European ancestry (EA) and AD in African ancestry (AA) comparing the proportion of significant genes with higher contribution weights of expression prediction models across all significant genes (*p* < 2.90 × 10^−6^). Black bars are the standard errors for the estimated proportions.

**Figure 5 ijms-26-02443-f005:**
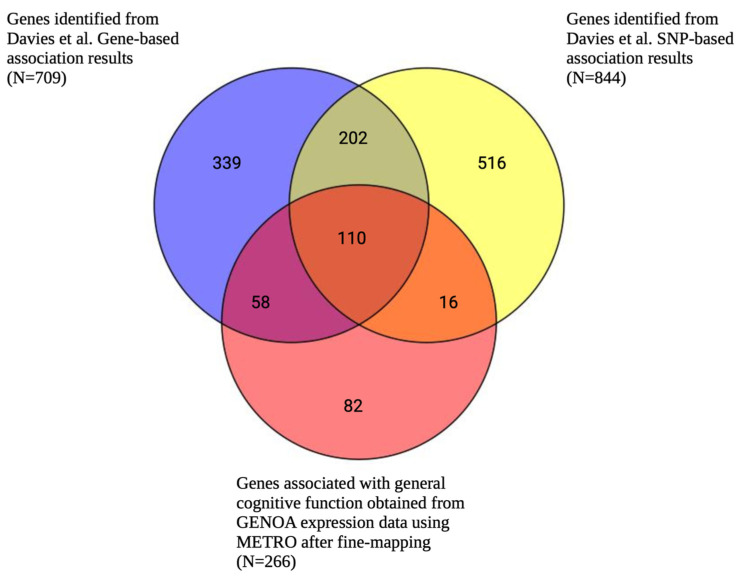
Venn diagram comparing number of METRO-identified genes associated with general cognitive function following FOCUS fine-mapping and genes identified by Davies et al. (2018) gene-based and SNP-based analyses. Venn diagram comparing the number of genes associated with general cognitive function obtained from METRO using GENOA gene expression data after Bonferroni correction (*p* < 2.90 × 10^−6^) and FOCUS [25] fine-mapping (red) and Davies et al. (2018) [21]. Davies et al. results included SNP-based association results that were mapped to the nearest gene (*p* < 5 × 10^−8^; yellow) and gene-based association results (*p* < 2.75 × 10^−6^; blue).

**Figure 6 ijms-26-02443-f006:**
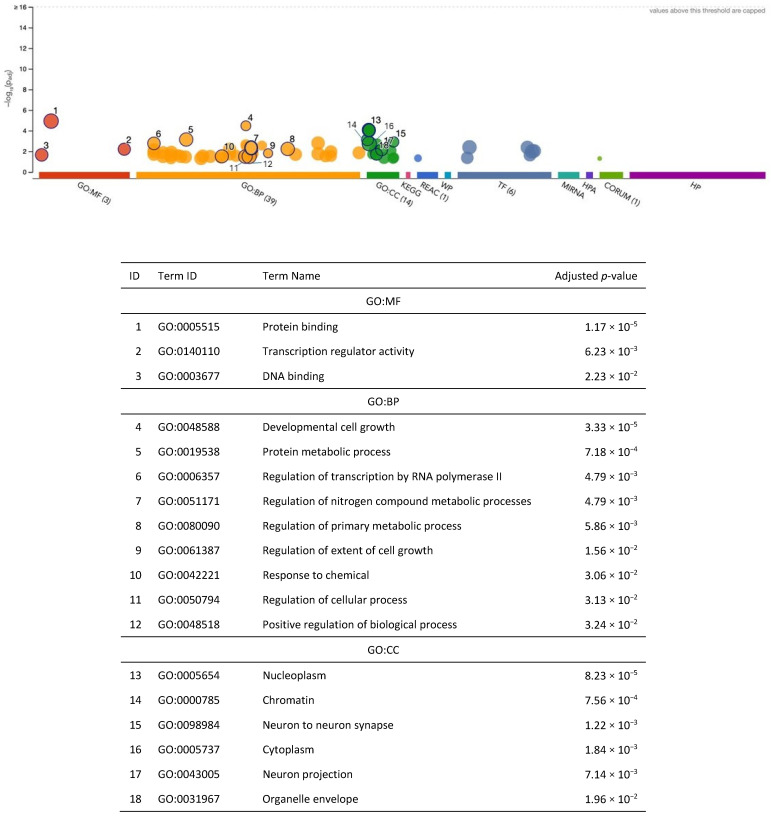
Functional enrichment analysis on the fine-mapped gene set identified for general cognitive function using METRO TWAS (N = 266 genes). The top panel consists of a Manhattan plot that illustrates the enrichment analysis results. The *x*-axis represents functional terms that are grouped and color-coded by data sources, including Gene Ontology (GO): molecular function (MF; red), GO: biological process (BP; orange), GO: cellular component (CC; dark green), Kyoto Encyclopedia of Genes and Genomes (KEGG; pink), Reactome (REAC; dark blue), WikiPathways (WP; turquoise), Transfac (TF; light blue), MiRTarBase (MIRNA; emerald green), Human Protein Atlas (HPA; dark purple), CORUM protein complexes (light green), and Human Phenotype Ontology (HP; violet), in order from left to right. The *y*-axis shows the adjusted enriched −log_10_ *p*-values < 0.05. Multiple testing correction was performed using the g:SCS method (Set Counts and Sizes) that takes into account overlapping terms. The top panel highlights driver GO terms identified using the greedy filtering algorithm in g:Profiler. The light circles represent terms that were not significant after filtering. The circle sizes are in accordance with the corresponding term size (i.e., larger terms have larger circles). The number in parentheses following the source name in the *x*-axis shows how many significantly enriched terms were from this source.

**Figure 7 ijms-26-02443-f007:**
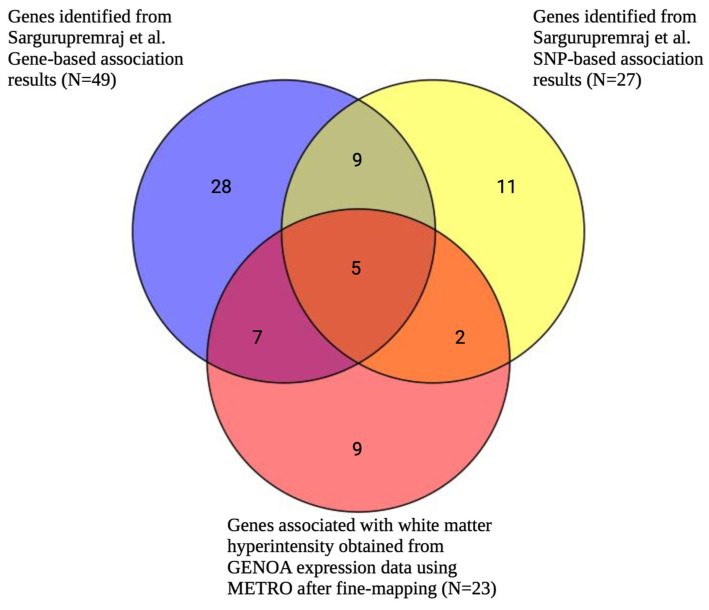
Venn diagram comparing number of METRO-identified genes associated with white matter hyperintensity following FOCUS fine-mapping and genes identified by Sargurupremraj et al. (2020) gene-based and SNP-based analyses. Venn diagram comparing the number of significantly associated genes associated with white matter hyperintensity (WMH) obtained from METRO using GENOA expression data after Bonferroni correction (*p* < 2.90 × 10^−6^) and fine-mapping (red) and Sargurupremraj et al. (2020) [22]. Sargurupremraj et al.’s results included SNP-based association results that were mapped to the nearest gene (*p* < 5 × 10^−8^; yellow) and gene-based association results (*p* < 2.77 × 10^−6^; blue).

**Figure 8 ijms-26-02443-f008:**
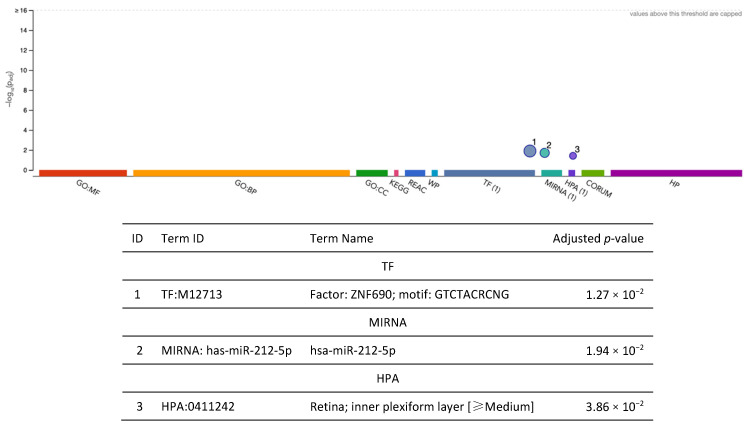
Functional enrichment analysis on the fine-mapped gene set identified for white matter hyperintensity using METRO TWAS (N = 23 genes). The top panel consists of a Manhattan plot that illustrates the enrichment analysis results. The *x*-axis represents functional terms that are grouped and color-coded by data sources, including Gene Ontology (GO): molecular function (MF; red), GO: biological process (BP; orange), GO: cellular component (CC; dark green), Kyoto Encyclopedia of Genes and Genomes (KEGG; pink), Reactome (REAC; dark blue), WikiPathways (WP; turquoise), Transfac (TF; light blue), MiRTarBase (MIRNA; emerald green), Human Protein Atlas (HPA; dark purple), CORUM protein complexes (light green), and Human Phenotype Ontology (HP; violet), in order from left to right. The *y*-axis shows the adjusted enriched −log_10_ *p*-values < 0.05. Multiple testing correction was performed using the g:SCS method (Set Counts and Sizes) that takes into account overlapping terms. The top panel highlights driver GO terms identified using the greedy filtering algorithm in g:Profiler. The light circles represent terms that were not significant after filtering. The circle sizes are in accordance with the corresponding term size (i.e., larger terms have larger circles). The number in parentheses following the source name in the *x*-axis shows how many significantly enriched terms were from this source.

**Figure 9 ijms-26-02443-f009:**
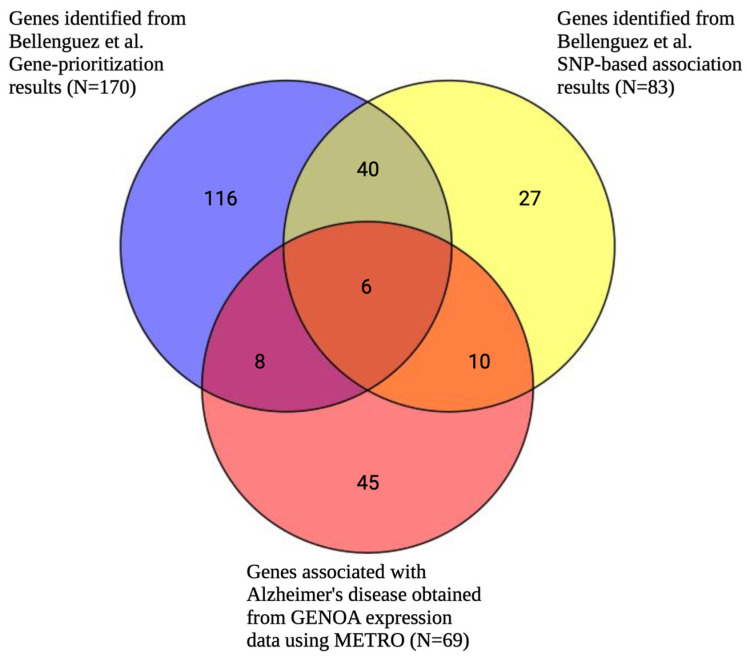
Venn diagram comparing number of METRO-identified genes associated with Alzheimer’s disease (EA GWAS) following FOCUS fine-mapping and genes identified by Bellenguez et al. (2022) gene prioritization and SNP-based analyses. Venn diagram comparing the number of significantly associated genes associated with Alzheimer’s disease (from EA GWAS) obtained from METRO using GENOA expression data after Bonferroni correction (*p* < 2.90 × 10^−6^) and fine-mapping (red) and Bellenguez et al. (2022) [23]. Bellenguez et al.’s results included SNP-based association results that were mapped to the nearest gene (*p* < 5 × 10^−8^; yellow) and gene prioritization results for the genes in the novel AD risk loci (blue). In the gene prioritization analysis, Bellenguez et al. analyzed the downstream effects of new AD-associated loci on molecular phenotypes (i.e., expression, splicing, protein expression, methylation, and histone acetylation) in various *cis*-quantitative trait loci (*cis*-QTL) catalogues from AD-relevant tissues, cell types, and brain regions.

**Figure 10 ijms-26-02443-f010:**
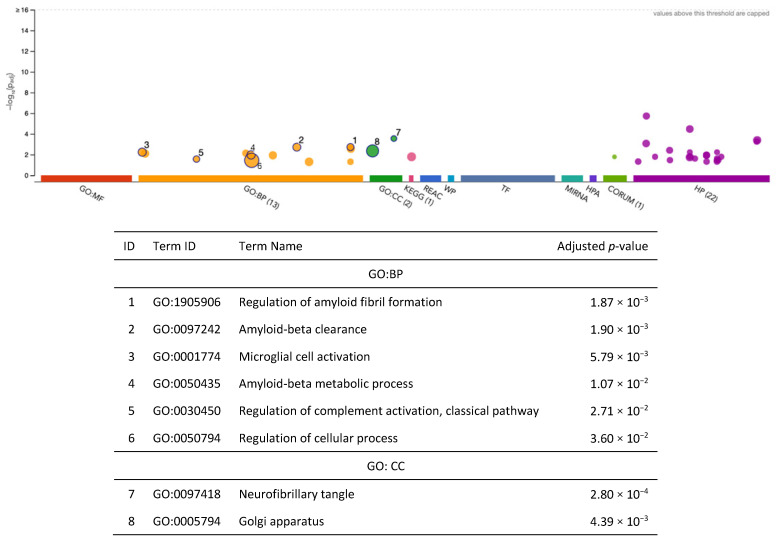
Functional enrichment analysis on the fine-mapped gene set identified for Alzheimer’s disease (EA GWAS) using METRO TWAS (N = 69 genes). The top panel consists of a Manhattan plot that illustrates the enrichment analysis results. The *x*-axis represents functional terms that are grouped and color-coded by data sources, including Gene Ontology (GO): molecular function (MF; red), GO: biological process (BP; orange), GO: cellular component (CC; dark green), Kyoto Encyclopedia of Genes and Genomes (KEGG; pink), Reactome (REAC; dark blue), WikiPathways (WP; turquoise), Transfac (TF; light blue), MiRTarBase (MIRNA; emerald green), Human Protein Atlas (HPA; dark purple), CORUM protein complexes (light green), and Human Phenotype Ontology (HP; violet), in order from left to right. The *y*-axis shows the adjusted enriched −log_10_ *p*-values < 0.05. Multiple testing correction was performed using the g:SCS method (Set Counts and Sizes) that takes into account overlapping terms. The top panel highlights driver GO terms identified using the greedy filtering algorithm in g:Profiler. The light circles represent terms that were not significant after filtering. The circle sizes are in accordance with the corresponding term size (i.e., larger terms have larger circles). The number in parentheses following the source name in the *x*-axis shows how many significantly enriched terms were from this source.

**Figure 11 ijms-26-02443-f011:**
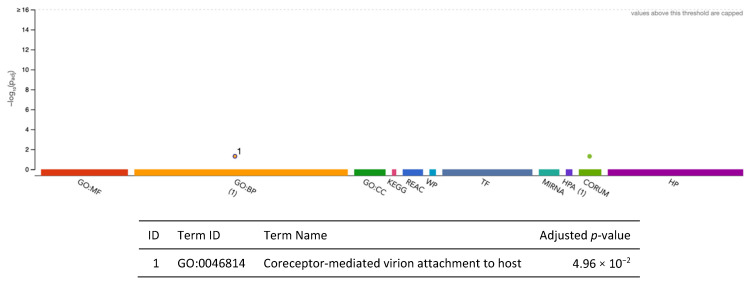
Functional enrichment analysis fine-mapped gene set identified for Alzheimer’s disease (AA GWAS) using METRO TWAS (N = 2 genes). The top panel consists of a Manhattan plot that illustrates the enrichment analysis results. The *x*-axis represents functional terms that are grouped and color-coded by data sources, including Gene Ontology (GO): molecular function (MF; red), GO: biological process (BP; orange), GO: cellular component (CC; dark green), Kyoto Encyclopedia of Genes and Genomes (KEGG; pink), Reactome (REAC; dark blue), WikiPathways (WP; turquoise), Transfac (TF; light blue), MiRTarBase (MIRNA; emerald green), Human Protein Atlas (HPA; dark purple), CORUM protein complexes (light green), and Human Phenotype Ontology (HP; violet), in order from left to right. The *y*-axis shows the adjusted enriched −log_10_ *p*-values < 0.05. Multiple testing correction was performed using the g:SCS method (Set Counts and Sizes) that takes into account overlapping terms. The top panel highlights driver GO terms identified using the greedy filtering algorithm in g:Profiler. The light circles represent terms that were not significant after filtering. The circle sizes are in accordance with the corresponding term size (i.e., larger terms have larger circles). The number in parentheses following the source name in the *x*-axis shows how many significantly enriched terms were from this source.

**Figure 12 ijms-26-02443-f012:**
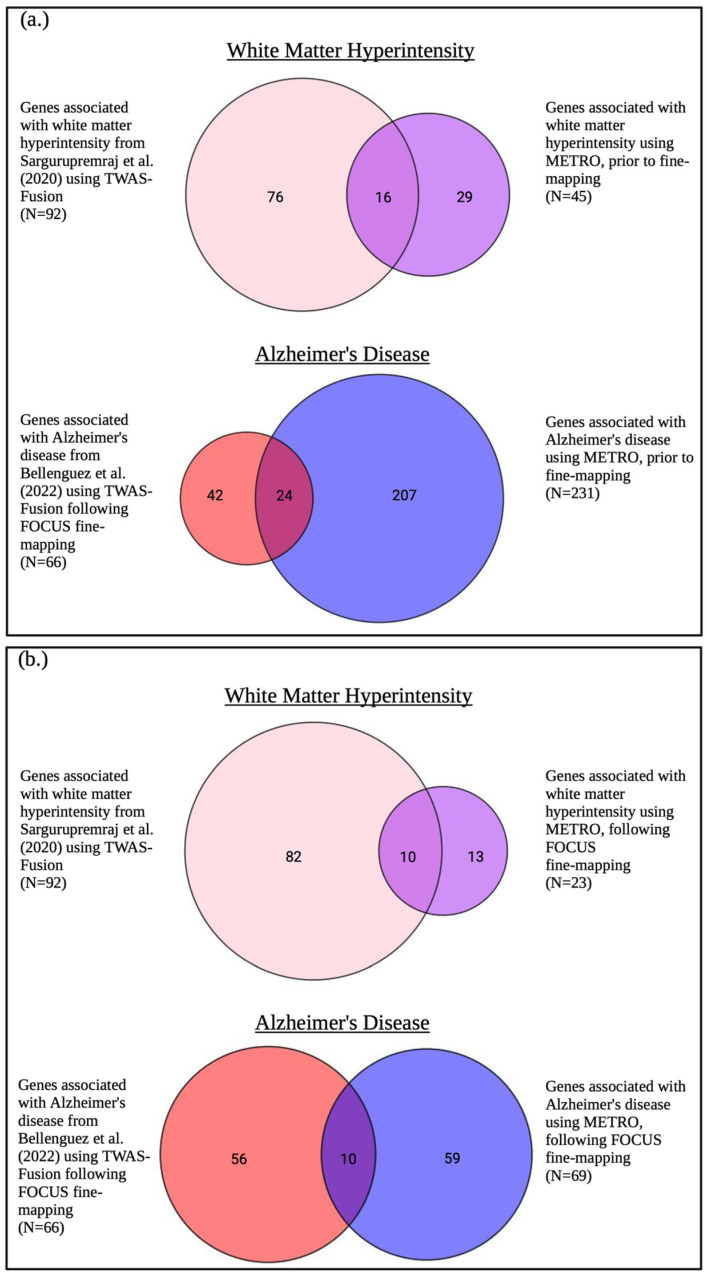
Venn diagram comparing METRO TWAS results prior to and following FOCUS fine-mapping with TWAS results from Sargurupremraj et al. (2020) and Bellenguez et al. (2022). Venn diagram comparing METRO TWAS results (**a**) prior to and (**b**) following FOCUS [25] fine-mapping with TWAS results using Fusion for white matter hyperintensity from Sargurupremraj et al. [22] (2020) without fine-mapping and Alzheimer’s disease from Bellenguez et al. [23] (2022) (EA GWAS) with FOCUS fine-mapping.

**Table 1 ijms-26-02443-t001:** Sample characteristics of expression quantitative trait locus (eQTL) mapping study and genome-wide association study (GWAS) participants.

** *eQTL mapping study: Genetic Epidemiology Network of Arteriopathy (GENOA)* **		
**Mean (SD) or N (%) or N**		
N		1833
Age (years)		56.85 (10.0)
Female		1202 (65.6%)
Race/Ethnicity		
	African Americans	1032 (56.3%)
	European Americans	801 (43.7%)
** *General cognitive function GWAS: CHARGE, COGENT, UKB ^a^* **		
**Mean (SD) or N (%) or N**		
N		300,486
Age (years)		56.91 (7.8)
Female		156,854 (52.2%)
Excluded for dementia and/or stroke diagnosis (N)		4919
** *White matter hyperintensity (WMH) GWAS: CHARGE and UKB ^a^* **		
**Mean (SD) or N (%) or N**		
N		48,454
Age (years)		64.17
Female		29,215 (57.6%)
WMH volume (cm^3^)		7.06 (8.8)
Excluded for stroke or pathologies (N)		1572
** *EA Alzheimer’s Disease (AD) GWAS: EADB, GR@ACE, EADI, GERAD/PERADES, DemGene, Bonn, the Rotterdam study, the CCHS study, NxC, and the UKB ^a^* **		
**Mean (SD) or N (%) or N**		
Discovery sample		
	AD cases (N)	39,106
	AD proxy cases (N)	46,828
	Controls (N)	401,577
Age (years)		
	AD cases or proxy cases	73.55 (8.1)
	Controls	67.86 (8.6)
Female		
	AD cases or proxy cases (N)	54,052 (62.9%)
	Controls (N)	48,209 (56.1%)
** *AA Alzheimer’s Disease (AD) GWAS: AD Genetics Consortium ^b^* **		
		**Mean (SD) or N (%) or N**
N		7970
	AD cases	2748 (34.5%)
	Controls	5222 (65.5%)
Age (years)		74.2 (13.6)
Female		
	AD cases	1944 (69.8%)
	Controls	3743 (71.7%)

Abbreviations: EA, European ancestry; AA, African ancestry. *^a^* GWASs [21,22,23] include only European ancestry participants. *^b^* GWAS [24] includes only African ancestry participants.

**Table 2 ijms-26-02443-t002:** Genes for WMH identified both by METRO followed by fine-mapping with FOCUS and by TWAS-Fusion conducted by Sargurupremraj et al. (2020) [22].

Gene	ENSG	Chr	Start	End	Gene Name	Accession Number (HGNC ID)
*CALCRL*	ENSG00000064989	2	188206691	188313187	Calcitonin receptor like receptor	HGNC:16709
*DCAKD*	ENSG00000172992	17	43100706	43138499	Dephospho-CoA kinase domain containing	HGNC:26238
*EFEMP1*	ENSG00000115380	2	56093102	56151274	EGF containing fibulin extracellular matrix protein 1	HGNC:3218
*GJC1*	ENSG00000182963	17	42875816	42908184	Gap junction protein gamma 1	HGNC:4280
*ICA1L*	ENSG00000163596	2	203637873	203736489	Islet cell autoantigen 1 like	HGNC:14442
*KLHL24*	ENSG00000114796	3	183353398	183402307	Kelch like family member 24	HGNC:25947
*NBEAL1*	ENSG00000144426	2	203879331	204091101	Neurobeachin like 1	HGNC:20681
*NEURL*	ENSG00000107954	10	105253462	105352303	Neuralized E3 ubiquitin protein ligase 1	HGNC:7761
*NMT1*	ENSG00000136448	17	43035360	43186384	N-myristoyltransferase 1	HGNC:7857
*WBP2*	ENSG00000132471	17	73841780	73852588	WW domain binding protein 2	HGNC:12738

Abbreviations: HGNC, Human Genome Organisation Gene Nomenclature Committee.

**Table 3 ijms-26-02443-t003:** Genes for AD identified both by METRO followed by fine-mapping with FOCUS and by TWAS-Fusion followed by fine-mapping with FOCUS conducted by Bellenguez et al. (2022) [23].

Gene	ENSG	Chr	Start	End	Gene Name	Accession Number (HGNC ID)
*BLNK*	ENSG00000095585	10	97948927	98031344	B cell linker	HGNC:14211
*CPSF3*	ENSG00000119203	2	9563780	9613230	Cleavage and polyadenylation specific factor 3	HGNC:2326
*DDX54*	ENSG00000123064	12	113594978	113623284	DEAD-box helicase 54	HGNC:20084
*GRN*	ENSG00000030582	17	42422614	42430474	Granulin precursor	HGNC:4601
*ICA1L*	ENSG00000163596	2	203637873	203736489	Islet cell autoantigen 1 like	HGNC:14442
*KLF16*	ENSG00000129911	19	1852398	1863578	KLF transcription factor 16	HGNC:16857
*LACTB*	ENSG00000103642	15	63414032	63434260	Lactamase beta	HGNC:16468
*PPP4C*	ENSG00000149923	16	30087299	30096697	Protein phosphatase 4 catalytic subunit	HGNC:9319
*SHARPIN*	ENSG00000179526	8	145153536	145163027	SHANK associated RH domain interactor	HGNC:25321
*TBX6*	ENSG00000149922	16	30097114	30103245	T-box transcription factor 6	HGNC:11605

Abbreviations: HGNC, Human Genome Organisation Gene Nomenclature Committee.

## Data Availability

The phenotype data and *APOE* genotypes used in the current study are available upon reasonable request to J.A.S. and S.L.R.K., and with a completed data use agreement (DUA). All other genotype data are available from the Database of Genotypes and Phenotypes (dbGaP): phs001401.v2.p1. Gene expression data are available from the Gene Expression Omnibus (GEO): GSE210256 and GSE138914. Due to IRB restriction, mapping of the sample IDs between genotype data (dbGaP) data (GEO) cannot be provided publicly, but is available upon written request to J.A.S. and S.L.R.K.

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
