# Peer review of "Multi-Ancestry Transcriptome-Wide Association Studies of Cognitive Function, White Matter Hyperintensity, and Alzheimer’s Disease"

_ijms, 2025, doi:10.3390/ijms26062443_

Round 1
Reviewer 1 Report
Comments and Suggestions for Authors
The manuscript presents a transcriptome-wide association study (TWAS) incorporating multi-ancestry datasets to investigate genetic associations with cognitive function, white matter hyperintensity (WMH), and Alzheimer’s disease (AD). The authors employ the METRO framework to integrate transcriptomic data from populations of different ancestries, addressing an important gap in genetic research that has historically been biased towards European cohorts. The study provides valuable insights into potential genetic factors underlying neurodegenerative conditions.
Major Comments
1. The manuscript employs the METRO framework to integrate multi-ancestry transcriptomic data, which is a valuable approach for enhancing genetic discovery beyond European-ancestry populations. However, the authors do not provide a clear justification for why METRO was chosen over other ancestry-aware transcriptomic imputation techniques, such as Bayesian models or ancestry-specific TWAS frameworks. Provide a comparative discussion of METRO against other available approaches, highlighting its advantages and any potential limitations in this specific context. A brief sensitivity analysis using an alternative model, if feasible, would further strengthen the argument for METRO’s selection.
2. While the study reports associations between various genes and cognitive function, WMH, and AD, it does not establish whether these associations are causal or simply statistical correlations. Consider incorporating Mendelian Randomization analyses or other causal inference techniques to determine whether gene expression changes play a direct role in AD pathogenesis. If MR is not possible due to data constraints, this limitation should be acknowledged, and alternative approaches discussed.
3. A growing body of literature suggests that metabolic dysfunction plays a key role in the onset and progression of AD. Insulin signaling, in particular, has been implicated in synaptic dysfunction, neuroinflammation, and amyloid pathology. Given these well-established links, the study would benefit from an exploration of whether insulin-related genes or pathways are significantly associated with cognitive function or AD within the dataset. The authors should assess whether their TWAS results include any statistically significant associations in genes related to insulin signaling or broader metabolic pathways.
4. While the study acknowledges ancestry-dependent differences in gene expression, it does not sufficiently explore the potential influence of environmental, socio-economic, or epigenetic factors on these differences. Given the well-documented disparities in AD prevalence across populations, discussing the interplay between genetic predisposition and non-genetic factors would enhance the interpretation of results. Include a discussion on how non-genetic factors might interact with ancestry-specific gene expression differences and whether any efforts were made to control for potential confounders in the analysis.
5. The study identifies several genes associated with neurocognitive traits, but it does not explore their functional relevance. While experimental validation may be beyond the scope of the paper, referencing studies that functionally characterize these genes in neurodegeneration would improve the biological interpretation of the findings. Include a brief discussion on whether any of the identified genes have been functionally validated in previous AD models, highlighting their potential roles in neurodegeneration.
Minor Comments
1. The manuscript states that different expression datasets were used, but it does not clearly describe how normalization was handled between platforms (e.g., Affymetrix Human Transcriptome Array 2.0 vs. Affymetrix Human Exon 1.0 ST Array). Clarify whether batch correction methods, such as ComBat or quantile normalization, were applied to harmonize expression data.
2. The manuscript alternates between "transcriptomic analysis" and "gene expression analysis," which may cause confusion. Define the terms explicitly and ensure consistent usage throughout the text.
3. Some statistical findings are reported with p-values but lack effect size information. Where relevant, include effect sizes or confidence intervals to provide a more complete picture of the magnitude of associations.
4. Gene names should follow standard conventions (italicized for human genes, all caps for mouse genes). Review and ensure consistency in gene formatting throughout the manuscript.
Reviewer 2 Report
Comments and Suggestions for Authors
This study uses multi-ancestry transcriptome-wide association studies (TWAS) to identify genes associated with cognitive function, white matter hyperintensity (WMH), and Alzheimer’s disease (AD). Utilizing gene expression data from both European ancestry (EA) and African ancestry (AA) samples, the researchers identified multiple genes, including ICA1L, which was associated with overlapping AD and vascular dementia (VaD) neuropathology. Enrichment analysis revealed involvement of pathways related to innate immunity, vascular dysfunction, and neuroinflammation. This work is the first TWAS of cognitive function and neurocognitive disorders leveraging multi-ancestry expression mapping, potentially expanding therapeutic targets for dementia.
ï‚·Gene Prioritization and Replication: Among the genes identified as significant in this study, were any subjected to independent replication or validation using additional datasets? If so, how do the findings compare, and if not, could the authors discuss the limitations this might pose?
Population-Specific Findings: The study highlights differences in gene associations between European and African ancestries. Could the authors provide further insights into whether these differences might be attributed to biological mechanisms, environmental factors, or statistical power differences due to sample sizes?
Clinical Relevance: The study identifies ICA1L as potentially contributing to overlapping AD and VaD neuropathology. Have the authors considered how this finding could be translated into clinical practice, such as biomarkers for early detection or therapeutic interventions? If not, could they elaborate on future steps to assess its clinical utility?
Recommended Citation: The readers must be interested in the relationship between WMH, VaD, and BBB function. Some papers found these relationships, which may be related to pathophysiology of the authors’ hypothesis. Please discuss this matter with reference to the following papers:
10.1186/s12987-023-00464-x
10.3389/fnagi.2023.1111448
10.14336/ad.2018.0929
10.1136/jnnp-2021-328519
Round 2
Reviewer 2 Report
Comments and Suggestions for Authors
The authors addressed my concerns.
Comments on the Quality of English LanguageGood